# Task Priors: Enhancing Model Evaluation by Considering the Entire Space of Downstream Tasks

## Abstract

A long-standing research problem in Artificial Intelligence (AI) is to produce systems that can successfully solve *any* possible task. A key requirement in addressing progress in that direction is a near-infinite suite of tasks for benchmarking AI solutions. In contrast, current evaluation methods available to AI researchers in representation learning typically rely on a fixed collection of hand-picked downstream benchmarks. Hence, a large amount of effort is put into designing and searching for large collection of evaluation tasks that can serve as a proxy of our grand goal. We argue that such a rigid evaluation protocol creates a structural bottleneck in AI research. To remedy that, we define a probability distribution over downstream tasks – **Task Priors**. Under this view, one can evaluate a model's performance over the set of all possible downstream tasks. Our framework is the first to provide answers to key questions such as *(i) what is the average performance of my model over all possible downstream tasks weighted by the probability to encounter each task?* or *(ii) what is the variance of my model's performance across all downstream tasks under the defined Task Priors?* Beyond establishing a new standard for evaluation, we believe that **Task Priors** will accelerate the pace of research in representation learning – where downstream task evaluation is generally the sole signal that researchers have access to.

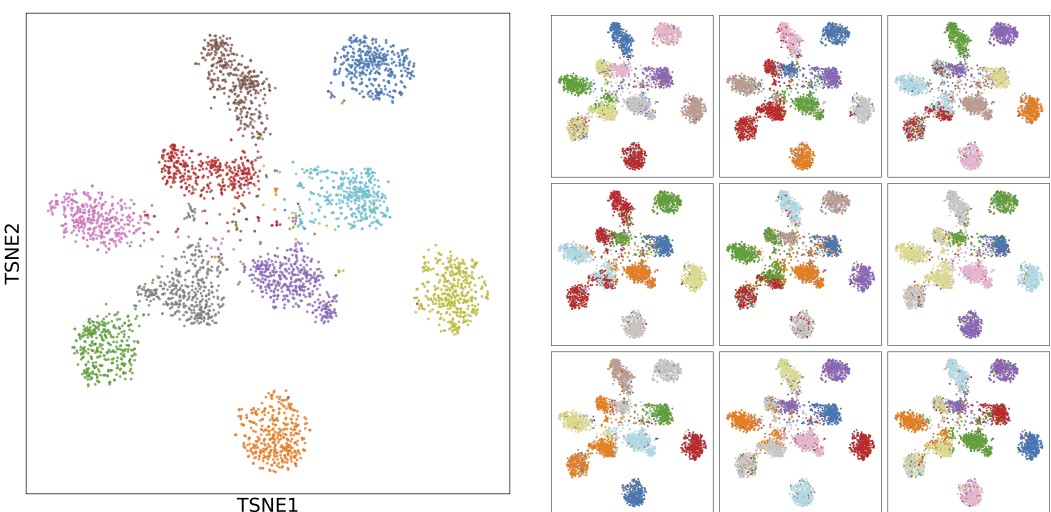

Figure 1: Comparison of the naive way to evaluate a model, only on the specific choice of labels provided with the Imagenette (Howard, 2019) dataset (Left) with the probabilistic view of targets sampled from the Task Prior, giving us a distribution we can evaluate on (Right). Colors here represent class labels.

# 1 INTRODUCTION

Pretrained backbone models today are released as a single checkpoint, yet in practice they can power millions of distinct downstream tasks: from simple classification and retrieval services to large-scale recommendation, autonomous perception, and more. On HuggingFace alone, top models such as *mobilenetv3*, *clip-vit*, and *bert* receive over 100 million downloads per month (Wightman, 2019). As the number of users and the diversity of applications grows, the space of possible downstream tasks that users want their models to perform well on tends toward infinity.

Yet, our standard evaluation protocols remain tethered to a small, fixed suite of hand-picked benchmarks, often around less than ten datasets (e.g., ImageNet, COCO, GLUE, SuperGLUE, WMT) (Lhoest et al., 2021; Russakovsky et al., 2015; Lin et al., 2014; Wang et al., 2018). Each new benchmark can take months of expert labeling and tens or even hundreds of thousands of dollars to assemble. Even large-scale benchmark suites that aggregate many tasks, such as the Massive Text Embedding Benchmark (MTEB), which spans 56 different evaluation datasets, or the similar Massive Image Embedding Benchmark (MIEB), still represent only a narrow slice of the possible space of downstream tasks (Muennighoff et al., 2023; Xiao et al., 2025). Once built, a static benchmark can only ever probe a tiny corner of the real-world tasks for which a model might be deployed. This disconnect creates a structural bottleneck between the handful of evaluation suites we all agree to evaluate our models on and the effectively infinite variety of tasks practitioners use our models for.

One way to break this bottleneck is simply to keep spending more time and money on ever-larger benchmarks, but that approach quickly becomes unsustainable. Instead, we propose a proxy evaluation framework that treats downstream tasks as samples from a well-defined probability space. By adopting a "*Task Prior*", a distribution over all possible targets informed by a feature kernel, we can compute expectations and variances of a model's downstream performance in closed form, without training new classifiers or designing new benchmarks. We show that this approach provides principled measures of average performance, robustness, and worst-case error across the entire landscape of potential tasks.

Our primary novel contributions in this work are as follows:

- **Task Priors.** We motivate the definition of the Task Prior by first proving that performance on linear probes is equivalent to maximizing the alignment of a kernel with the induced label graph, unifying supervised and self-supervised evaluation. We then formalize the space of downstream tasks as a Gibbs distribution over all possible label graphs.

- **Closed-form Statistics.** From the Task Prior we derive $O(n^2)$ formulas for the expected downstream error and its variance, bypassing benchmark curation or probe training.

- **Efficient Sampling from the Task Prior.** We introduce an $O(n)$ prefix-sampling algorithm that draws realistic classification tasks from the prior, enabling cheap evaluation on many tasks.

- **Empirical Validation.** We find that across a range of backbones, our closed form metrics on kernel align with mean and variance of accuracy of linear probes on a subset of ImageNet. We then show that the performance implied by our Task Prior correlates with performance on a hand curated set of downstream tasks.

# 2 TASK PRIORS: MODEL EVALUATION ON INFINITELY MANY TASKS

This section bridges supervised "absolute" objectives and pairwise "relative" objectives and uses that link to define a Task Prior tied to our pretrained feature kernels. Concretely, we prove the loss equivalence (2.1), introduce a Gibbs prior over label graphs (2.2), connect it to SSL formulations (2.3), and present a fast sampler for evaluation of probes (2.4). In this work, we elect to focus on *classification tasks,* as we believe this to be a sufficiently rich primitive for evaluation.

## 2.1 AN EQUIVALENCE BETWEEN LOSSES

Traditional downstream evaluation relies on *absolute objectives*, and asks, "How well can a linear head map features back to a specific label vector $\mathbf{Y}$?". This works when a single, human-defined

labeling is the goal, but it scales poorly once we care about many downstream tasks. By recasting supervised objectives in the language of pairwise relations by looking at the sample-sample adjacency matrix $\mathbf{G}$, we can shift to look at the more general setting of *relative objectives*. In this section, we will relate these two types of objectives and derive a natural equivalence between supervised and self-supervised objectives. In the sequel, we will leverage this connection to define our Task Prior. To better explain our findings, we first need to introduce the formal definitions of *relative* and *absolute* objectives.

**Definition 2.1** (Absolute objective). *An **absolute** objective compares a prediction of sample $\mathbf{x}_n$ to the target $\mathbf{y}_n$ independently from the other samples.*

**Definition 2.2** (Relative objective). *A **relative** objective compares a $N \times N$ inter-sample relation matrix (e.g., $f_\theta(\mathbf{X})^\top f_\theta(\mathbf{X})$) to a $N \times N$ inter-label relation matrix $\mathbf{G}$ (e.g. $\mathbf{G} = \mathbf{Y}^\top \mathbf{Y}$).*

In many practical settings, rather than focusing solely on self-supervised loss formulations, we are given a pretrained representation $f_\theta(\mathbf{X})$ and wish to assess how well a simple linear head can recover downstream tasks. By solving the unconstrained mean squared error objective for the linear probe, we find that the optimal loss depends only on the kernel matrix $\mathbf{M} = f_\theta(\mathbf{X})^\top f_\theta(\mathbf{X})$ and the label adjacency graph $\mathbf{G} = \mathbf{Y}^\top \mathbf{Y}$:

Let's denote the input data as the matrix $\mathbf{X} \triangleq [\mathbf{x}_1, \ldots, \mathbf{x}_N]$ the targets or labels as the matrix $\mathbf{Y} \triangleq [\mathbf{y}_1, \ldots, \mathbf{y}_N]$ where for classification each $\mathbf{y}_n$ is a one-hot vector at the corresponding class, and the model's prediction as $f_\theta(\mathbf{X}) \triangleq [f_\theta(\mathbf{x}_1), \ldots, f_\theta(\mathbf{x}_N)]$. Finally, let's denote the $N$-dimensional centering matrix by $\mathbf{H} \triangleq \mathbf{I} - \frac{1}{N}\mathbf{1}_N\mathbf{1}_N^\top$.

**Unconstrained linear classifier.** We are now minimizing the usual supervised mean squared error between the affinely transformed backbone output $\mathbf{W}f_\theta(\mathbf{X}) + \mathbf{b}\mathbf{1}_N^\top$ and the targets $\mathbf{Y}$ as in

$$\mathcal{L}(\mathbf{W}, \mathbf{b}, \theta) \triangleq \frac{1}{N}\left\|\mathbf{W}f_\theta(\mathbf{X}) + \mathbf{b}\mathbf{1}_N^\top - \mathbf{Y}\right\|_F^2 \tag{1}$$

Our goal will be to minimize that loss with respect to $\mathbf{W}, \mathbf{b}$ and to perform some algebraic manipulations to demonstrate how that *absolute* objective–comparing the prediction of sample $n$ to the label of sample $n$–turns into a *relative* objective–comparing pairwise predictions to pairwise labels. The following derivations do not rely on any assumptions about the input $\mathbf{X}$, the network $f_\theta$, or the targets $\mathbf{Y}$, other then that the representations $f_\theta(\mathbf{X})$ are of full rank, which is generally true. We will also denote the Singular Value Decomposition of $f_\theta(\mathbf{X})$ as $\mathbf{U}\mathbf{\Sigma}\mathbf{V}^\top$.

**Theorem 2.3.** *The optimum of eq.* (1) *w.r.t.* $\mathbf{W}, \mathbf{b}$ *can be obtained in closed-form as*

$$\min_{\mathbf{W}, \mathbf{b}, \theta} \mathcal{L}(\mathbf{W}, \mathbf{b}, \theta) = \min_\theta -\frac{1}{N}\operatorname{Tr}\left(\mathbf{V}^\top\mathbf{Y}^\top\mathbf{Y}\mathbf{V}\right) + \text{cst}. \tag{2}$$

*(Proof in Appendix A.1.)*

This theorem shows that the absolute objective from Definition 2.1 is equivalent to the associated relative objective in Definition 2.2. A crucial benefit of Equation (2) is that it only requires $\mathbf{G}$, which could be built from $\mathbf{Y}$, or without labels at all. For example, in a classification setting, Definition 2.1 requires the categorical label for each sample, while Definition 2.2 only requires to know which samples are from the same class *but not what that class is*.

In the *relative* objective, one no longer tries to predict $\mathbf{y}_n$ from $\mathbf{x}_n$, but instead tries to make the pairwise comparison of predictions $\mathbf{V}\mathbf{V}^\top$ aligned with the pairwise comparison of labels $\mathbf{Y}^\top\mathbf{Y}$. In particular, we clearly see that the left singular vectors of $f_\theta(\mathbf{X})$ no longer contribute to the loss, as expected since the optimal $\mathbf{W}$ automatically maps them to the left singular vectors of $\mathbf{Y}$. Hence, we are left with the right singular vectors $\mathbf{V}$ of $f_\theta(\mathbf{X})$. Notice that the trace on the right hand side of (2) is not trivial, as $\mathbf{V}$ denotes the truncated SVD, and has orthogonal columns but does not necessarily have orthogonal rows. This is true in the typical case when the number of data points exceeds the feature dimension. So, since we can regard $\mathbf{Y}^\top\mathbf{Y} = \mathbf{G}$ and $\mathbf{V}\mathbf{V}^\top$ is a valid kernel we can write as $\mathbf{K}$, then we can write our trace as $-\frac{1}{N}\operatorname{Tr}(\mathbf{G}\mathbf{K})$.

## 2.2 A Distribution Over Target Labels

Building on Theorem 2.3, we can now introduce a prior distribution over label graphs $\mathbf{G}$ that reflects the likelihood of different downstream tasks. To achieve this, we define a measure over all possible

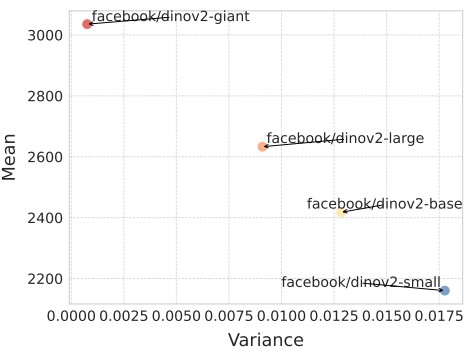 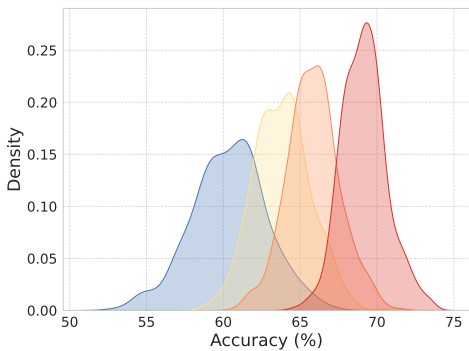

Figure 2: Here we show an example of using Task Priors to evaluate the DinoV2 (Oquab et al., 2023) family of models, with respect the the Task Prior kernel, *dinov2-giant*. We show the expectation and variance of $\text{Tr}(\mathbf{KG})$ (Left), as well as the distribution of performance of linear probes on labelings sampled from the Task Prior (Right).

$\mathbf{G}$ matrices, weighted by alignment with a pretrained feature kernel $\mathbf{K}$. This will assign a higher probability to more realistic scenarios that might be encountered in real-world applications. This *Task Prior* allows us to compute expectations and variances of kernel alignment scores in closed form, and to efficiently sample realistic downstream targets without training additional classifiers.

To begin, suppose that we have a kernel function that measures similarity between data points $k$ : $\mathbb{R}^D \times \mathbb{R}^D \to \mathbb{R}$. Let $\mathbf{K}$ then be a kernel matrix corresponding to $\mathbf{X}$. For the rest of the paper we can assume this is the centered kernel matrix corresponding to the features. Now if we have a graph adjacency matrix $\mathbf{G}$, we can read off the elements of the product, $[\mathbf{GK}]_{ij} = \sum_{k=1}^{N} \mathbf{G}_{ik} \, k(\mathbf{x}_j, \mathbf{x}_k)$. In particular we can see that the diagonal elements of this matrix is given as follows, where we use $\mathbf{x}_j \sim \mathbf{x}_i$ to mean that $\mathbf{x}_j$ and $\mathbf{x}_i$ are connected in graph $\mathbf{G}$, $[\mathbf{GK}]_{ii} = \sum_{k=1}^{N} \mathbf{G}_{ik} \, k(\mathbf{x}_i, \mathbf{x}_k) = \sum_{\mathbf{x}_k \sim \mathbf{x}_i} k(\mathbf{x}_k, \mathbf{x}_i)$. Summing over $i$ gives the trace,

$$\text{Tr}(\mathbf{GK}) = \sum_{i=1}^{N} \sum_{\mathbf{x}_k \sim \mathbf{x}_i} k(\mathbf{x}_i, \mathbf{x}_k),$$

which acts as a global *compatibility score* between the geometry in the label graph $\mathbf{G}$ and the kernel $\mathbf{K}$. We treat the negative trace, $\mathcal{E}(\mathbf{G}) := -\text{Tr}(\mathbf{GK})$, as the "energy" of a labeling: graphs that connect feature-similar points (high trace of $\mathbf{GK}$) have lower energy and are therefore more likely. This leads to the naturally to the Gibbs measure, $\mu(\mathbf{G}) \propto e^{-\mathcal{E}(\mathbf{G})/T} = e^{\text{Tr}(\mathbf{GK})/T}$, with temperature $T > 0$. As we increase the temperature, this distribution tends towards the uniform distribution on all graphs, and we expect to see more interesting behavior in the low temperature regime. The properties of this distribution enable direction computation of the expectation and higher moments, and enable efficient sampling algorithms which we develop in the sequel.

**Definition 2.4.** *(Task Prior Distribution) Given a kernel matrix $\mathbf{K}$ on $n$ data points, and a temperature $T > 0$, we will define the following Gibbs measure on the space of all graphs, $G$ as:*

$$\mu(\mathbf{G}) \propto e^{\frac{\text{Tr}(\mathbf{GK})}{T}}, \tag{3}$$

*where we denote by $Z_{T,\mathbf{K}}$ the corresponding partition function.*

If we have multiple task prior kernels, $\mathbf{K}_1, \mathbf{K}_2$, then we can compute that $\mu_{\mathbf{K}_1}(\mathbf{G})\mu_{\mathbf{K}_2}(\mathbf{G}) \propto \mu_{\mathbf{K}_1+\mathbf{K}_2}(\mathbf{G})$, giving an easy way to leverage multiple backbones for the task prior distribution.

Although computing exactly the probability of observing a single graph can be quite challenging, as computing the partition function would require $2^{N^2}$ computations, the specific structure of this probability measure admits a convenient factorization on a per-edge level.

**Lemma 2.5.** *Suppose that we consider the Gibbs measures over all graphs $\mathbf{G}$. Then, the probability of a single edge $i, j$ being present is given by,*

$$\mathbb{P}(\mathbf{G}_{i,j} = 1) = \sigma(\mathbf{K}_{i,j}/T), \tag{4}$$

*where $\sigma$ denotes the sigmoid function. Furthermore if $(i,j) \neq (l,k)$, then,*

$$\mathbb{P}(\mathbf{G}_{i,j} = 1 \;\wedge\; \mathbf{G}_{i,j} = 1) = \sigma(\mathbf{K}_{i,j}/T)\sigma(\mathbf{K}_{l,k}/T). \tag{5}$$

*(Proof in Appendix A.2.)*

The above lemma allows us to, given some kernel matrix driven by an assumption on similarity over our data points, $\mathbf{K}$, evaluate the performance of an representation model providing another kernel matrix $\mathbf{M}$.

**Theorem 2.6.** *Given a kernel matrix $\mathbf{K}$ and associated Gibbs measure $\mu_{\mathbf{K}}$, and another kernel matrix $\mathbf{M}$, we can compute the expectation of $\mathrm{Tr}(\mathbf{MG})$ as follows,*

$$\mathbb{E}_{\mathbf{G}\sim\mu_{\mathbf{K}}}\left[\mathrm{Tr}(\mathbf{MG})\right] = \sum_{1 \leq i,j \leq N} \mathbf{M}_{i,j}\mathbb{P}_{\mathbf{G}\sim\mu_{\mathbf{K}}}(\mathbf{G}_{i,j}=1) = \sum_{1 \leq i,j \leq N} \mathbf{M}_{i,j}\,\sigma(\mathbf{K}_{i,j}/T). \tag{6}$$

*Furthermore, the variance satisfies,*

$$\mathrm{Var}(\mathrm{Tr}(\mathbf{MG})) = \sum_{1 \leq i,j \leq N} \mathbf{M}_{i,j}^2\,\sigma(\mathbf{K}_{i,j}/T)(1 - \sigma(\mathbf{K}_{i,j}/T)).$$

*(Proof in Appendix A.3)*

We will note that computing the mean and variance of $\mathrm{Tr}(\mathbf{MG})$, when $\mathbf{G}$ is distributed according to the task prior, takes on the order of $O(N^2)$ computations for $N$ data points.

## 2.3 SSL OBJECTIVES CAN SERVE AS TASK PRIORS

If you don't have a prior kernel matrix, you may be able to use the same modeling assumptions you use in Self-Supervised Learning (SSL) to create an admissible kernel matrix (van den Oord et al., 2018; Bachman et al., 2019; He et al., 2020; Caron et al., 2020; Chen et al., 2020a;b; Henaff, 2020; Zbontar et al., 2021; Wang & Isola, 2020; Assran et al., 2023). To start, while the above section demonstrates that supervised and SSL abide by the same objective, i.e., minimizing the absolute loss eq. (1) with $\mathbf{Y}$ or minimizing the relative loss eq. (2) with $\mathbf{G} = \mathbf{Y}\mathbf{Y}^\top$ is equivalent, it remains to show what $\mathbf{G}$ SSL is actually employing.

We recall that in a typical SSL setting, each sample is augmented $V$ times to form $V$ views. Hence we now have $\mathbf{G} \in \mathbb{R}^{NV \times NV}$. And the loss aims at collapsing together the views of each sample, i.e., $(\mathbf{G})_{i,j} = 1_{\{\lfloor i/V \rfloor = \lfloor j/V \rfloor\}}$, assuming that the data samples are ordered based on their index first, and augmentations second. The question that naturally arises is about the underlying $\mathbf{Y}$ that the loss is considering. We formalize that result below.

**Proposition 2.7.** *The SSL graph given by $(\mathbf{G})_{i,j} = 1_{\{\lfloor i/V \rfloor = \lfloor j/V \rfloor\}}$ is recovered from considering the labels $\mathbf{Y} \in \mathbb{R}^{NV \times N}$ with $(\mathbf{Y})_{n,j} = 1_{\{\lfloor n/V \rfloor = 1\}}$.*

That is, SSL is implicitly acting on a labeling of the dataset that attributes to each sample a unique class. As such a labeling forces the model to maintain information about all its training samples–at least enough to separate them– it is clear that this is what maximized the Mutual Information between the original data $\mathbf{X}$ and the final learned representation $f_\theta(\mathbf{X})$ (Shwartz-Ziv & LeCun, 2023). So, we can use this *graph* $\mathbf{G}$ as the *kernel* matrix to be our task prior.

## 2.4 SAMPLING TASKS FOR EVALUATION

Starting from the task-prior distribution $\mu_{\mathbf{K}}$ over graphs introduced in the previous section, we can view every edge as an independent Bernoulli variable whose success probability is $\sigma(\mathbf{K}_{i,j}/T)$. However, for the the purpose of measuring performance of a model with linear probes, we may instead want to sample from the probability measure restricted on those graphs $\mathbf{G}$ which arise from one hot labelings $\mathbf{Y}$, where $\mathbf{G} = \mathbf{Y}\mathbf{Y}^\top$. We will denote by $\mu_{\mathbf{K}}^q$ the probability measure given by,

$$\mu_{\mathbf{K}}^q(G) \propto \begin{cases} \mu_{\mathbf{K}}(\mathbf{G}) & \text{if } \mathbf{G} = \mathbf{Y}\mathbf{Y}^\top \text{ for some one-hot } Y \in \{0,1\}^{N \times q} \\ 0 & \text{else} \end{cases}.$$

Sampling from the restricted measure $\mu_{\mathbf{K}}^q$ is a much more challenging problem, and is equivalent to sampling from the Potts Model from statistical mechanics (Wu, 1982). For instance, for binary

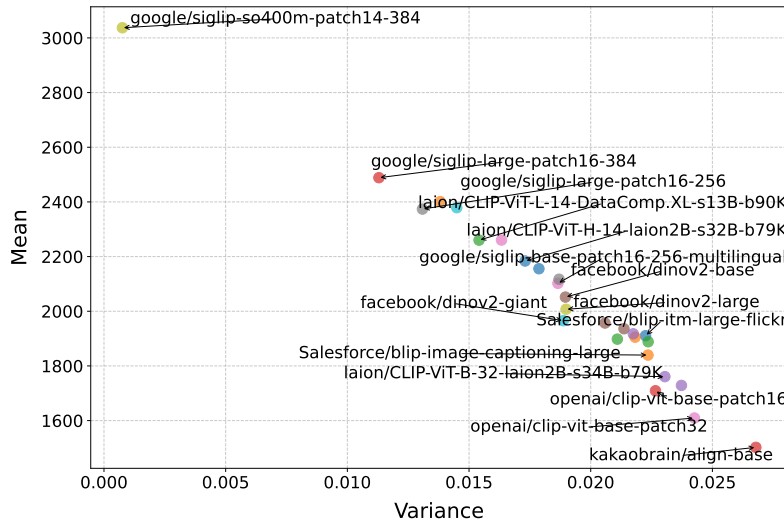

Figure 3: We plot the expectation and variance of $\text{Tr}(\mathbf{MG})$, where $\mathbf{M}$ is the centered cosine similarity kernel matrix for each models features generated from mini-imagenet (Russakovsky et al., 2015), where the expectation is taken against $\mu_{\mathbf{K}}$. Please see the appendix for more information and ablation on temperature and choice of prior kernel.

labelings there are $2^n$ possible states, so sampling and computing the partition function can be completely intractable. We could use Markov Chain Monte Carlo (MCMC) methods such as the Metropolis-Hastings algorithm to sample from this distribution, but this is sensitive to hyperparameters and may mix slowly in practice. Instead, we propose an approximate sampling algorithm in $O(n)$ time to sample a labeling on $n$ data points.

Suppose we write the labeling $\mathbf{Y} = [y_1, \ldots y_n]$, and we denote by $\mathbf{c}$ a one hot vector corresponding to class $c$. We operate sequentially, assigning a label to each new data point we see using the following approximation of the true measure $\mu_K^q$, where $p(y_i = \mathbf{c}|y_1, \ldots, y_{i-1}) \approx \frac{1}{C}\exp(\frac{1}{T}\sum_{j<i}\mathbf{K}_{i,j}\mathbf{1}_{\{y_j=\mathbf{c}\}})$. We can then achieve an algorithmic speedup by using the factorization of our kernel matrix as $\mathbf{K} = \mathbf{Z}\mathbf{Z}^T$ (if we do not have access to the features, we can use for instance a Cholesky factorization here). Then we have,

$$\exp(\frac{1}{T}\sum_{j<i}\mathbf{K}_{i,j}\mathbf{1}_{\{y_j=\mathbf{c}\}}) = \exp(\frac{1}{T}\mathbf{Z}_i\sum_{j<i}\mathbf{Z}_j\mathbf{1}_{\{y_j=\mathbf{c}\}}). \tag{7}$$

From equation 7, we can devise our method for the sampling Algorithm 1. Using this algorithm, we are able to quickly sample labelings of the data points according to the task prior, as demonstrated in Figure 1, where we apply this on the Imagenette dataset (Howard, 2019) to features generated by the pretrained *ResNet-18*. [1]

## 3 EMPIRICAL APPLICATIONS OF TASK PRIORS

This section applies Task Priors to empirical evaluation of representation learning models. Here, we assess model kernels directly (3.1), show these statistics predict linear-probe accuracy on sampled labels (3.2), and validate alignment with curated benchmarks (3.3).

---

[1]Interested readers can experiment with different class counts and temperature settings in this Colab notebook: https://colab.research.google.com/drive/1qNOgoNSH87AcdODug-yop7Q0MuT8w1r7

---

**Algorithm 1** Prefix Sampler for Multi-Class Task Prior

---

**Require:** $\mathbf{Z} \in \mathbb{R}^{N \times r}$          ▷ factor so $\mathbf{K} \approx \mathbf{Z}\mathbf{Z}^\top$
**Require:** $T > 0$          ▷ temperature
**Require:** $q \in \mathbb{N}$          ▷ number of classes
**Ensure:** $labels \in \{0, \ldots, q-1\}^N$
 1: allocate $labels[1{:}N]$
 2: $\mathbf{U} \leftarrow 0_{r \times q}$          ▷ class-wise prefix sums
 3: **for** $i = 1 \ldots N$ **do**
 4:      $h \leftarrow \left(\frac{1}{T}\right)(\mathbf{Z}[i,:] \, \mathbf{U})$          ▷ length $q$ vector
 5:      $h \leftarrow h - \max(h)$          ▷ stabilize
 6:      $p \leftarrow \exp(h);$    $p \leftarrow p / \sum p$
 7:      $c \sim \text{CategoricalSample}(p)$
 8:      $labels[i] \leftarrow c$
 9:      $\mathbf{U}[:,c] \mathrel{+}= \mathbf{Z}_{i,:}$
10: **end for**
11: **return** $labels$

---

## 3.1 EVALUATING MODEL KERNELS

We use the equations given in Theorem 2.6 as a way to evaluate model performance in a extremely efficient way, i.e. without training any probes, or even assembling a collection of tasks / benchmarks. We demonstrate this in Figure 3 and in Figure 2 (Left) on a selection of models trained to learn representations from images, on a subset of $8{,}192$ images from *mini-imagenet* (Russakovsky et al., 2015). We use the centered cosine similarity as the choice of kernel here and in the rest of the experiments in this paper. We also find that the mean and variance are negatively correlated, implying that models that perform well on average tend also to perform better across a variety of tasks.

## 3.2 TASK
PRIORS PREDICTS PERFORMANCE ON SAMPLED LABELS

A central claim of this paper is that the two kernel statistics, $\mathbb{E}_{\mu_{\mathbf{K}}}[\text{Tr}(\mathbf{M}\mathbf{G})]$ and $\text{Var}_{\mu_{\mathbf{K}}}(\text{Tr}(\mathbf{M}\mathbf{G}))$, can predict a representation's downstream performance as measured by linear probes. For Figures 4 and 2 (Right), we sample tasks from a prior $\mu_{\mathbf{K}}$ induced by *DinoV2-giant*. For Figure 4, we train an independent linear probe on every sampled task with and record the resulting accuracies. We then compare this to simply computing $\mathbb{E}_{\mu_{\mathbf{K}}}[\text{Tr}(\mathbf{M}\mathbf{G})]$ and $\text{Var}_{\mu_{\mathbf{K}}}(\text{Tr}(\mathbf{M}\mathbf{G}))$ as per Theorem 2.6. Concretely, for each of the models displayed in Figure 3, we compute the average and the variance of accuracies on linear probes, and compare them to the first and second moments, $\mathbb{E}_{\mu_{\mathbf{K}}}[\text{Tr}(\mathbf{M}\mathbf{G})]$ and $\text{Var}_{\mu_{\mathbf{K}}}(\text{Tr}(\mathbf{M}\mathbf{G}))$, computed with *DinoV2-giant* as a prior.

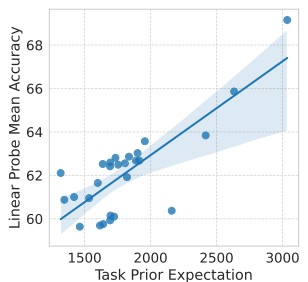

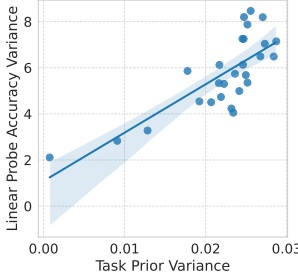

Figure 4: Correlation between the mean and variance of $\text{Tr}(\mathbf{G}\mathbf{K})$, and the accuracy of linear probes sampled by the same Task Prior. We observe a Spearman correlation of $0.68$ (top) and $0.76$ (bottom).

We note that the these statistics as shown in Figure 3, tends to exhibit the same trends as the model's linear probe performance on sampled tasks, as corroborated by Figure 8, where stronger models tend to have a higher average accuracy / trace, as well as a lower variance. We observe a strong correlation between these sets of data points in Figure 4.

## 3.3 TASK PRIORS ALIGN WITH PERFORMANCE ON CURATED BENCHMARKS

For this framework to be useful to practitioners, the performance on a hand-curated collection of downstream tasks should follow the distribution implied by the Task Prior. In the previous section

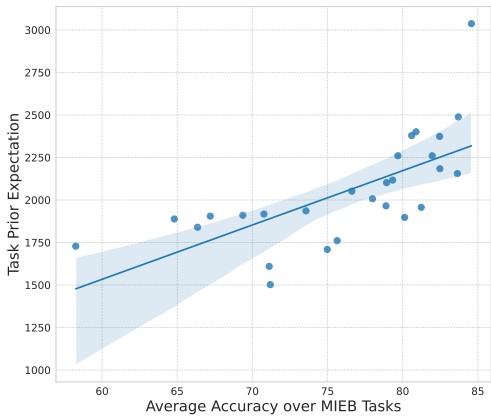 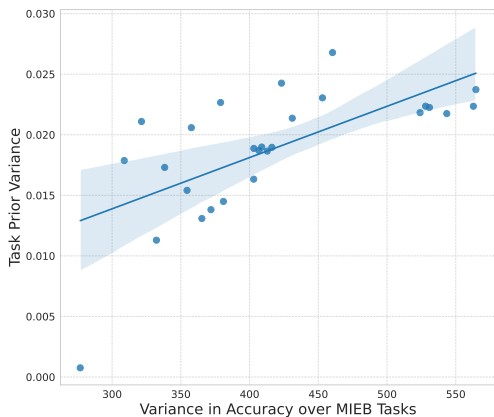

Figure 5: For each of the 26 models, we estimate the task-prior expectation and variance and compare them to that model's mean accuracy over a hand-curated set of 22 MIEB classification tasks (Xiao et al., 2025). We observe a Spearman correlation of $0.82$ (Left) and $0.74$ (Right), showcasing how Task Priors can accurately predict model performance on a distribution of tasks.

section, we established that these theoretic measures based on the trace of the matrix $\mathbf{GK}$ can represent performance on linear probes as generated via the task prior. In this section, we verify an analogous result but for a hand-curated collection of downstream tasks for image classification found in the MIEB (Xiao et al., 2025). We focus our analysis on a selection of 26 models easily available through *huggingface*. The MIEB paper Tables 16, 17 report the accuracy of each of these models on each of 22 downstream classification tasks. Using the strongest model on the MIEB tasks, *google/siglip-so400m-patch14-384*, as our task prior, we find that the mean and variance of the linear probe accuracy across these downstream tasks correlate to $\mathbb{E}_{\mu_{\mathbf{K}}}[\mathrm{Tr}(\mathbf{MG})]$ and $\mathrm{Var}_{\mu_{\mathbf{K}}}(\mathrm{Tr}(\mathbf{MG}))$ respectively, as demonstrated in Figure 5. In this sense, Task Priors is able to predict the distribution of a model's performance over real downstream tasks. We examine the bias incurred via the choice of prior model in Figure 6.

## 4 RELATED WORKS

Many works aim to capture the performance of representation models primarily by intrinsic quantities about the model's features. For instance, RANKME measures the *effective rank* of the feature matrix and shows a empirical correlation with average linear-probe accuracy across several tasks (Garrido et al., 2023). LIDAR argues that a method built on Linear Discriminant Analysis serves as a proxy for downstream performance (Thilak et al., 2023). More recent works take a broader view, demonstrating that k-NN, few-shot fine-tuning, and clustering evaluations may all disagree in systematic ways (Marks et al., 2025). Collectively, these studies show that properties intrinsic to the representation can forecast downstream success, but they still reduce performance to one or two scalar summaries.

A complementary line of work attacks the evaluation bottleneck by increasing the number of test tasks. In NLP, suites such as MTEB (56 embedding datasets) (Muennighoff et al., 2023) and HELM (42 scenarios, seven axes of measurement) (Liang et al., 2023) provide broad coverage of downstream tasks. The same trend is exists in vision, with works such as *VideoEval* packaging twenty diverse video understanding datasets together (Li et al., 2024), and frameworks such as MIEB (Xiao et al., 2025) providing a curated collection of downstream tasks for vision and multi-modal models.. While these large benchmarks can be quite helpful for practitioners, they remain *finite* and expensive to create. Worse, even a hundred benchmarks sample only a vanishingly small corner of the large task space practitioners can care about.

Our *Task Priors* framework can be viewed as the missing bridge between these two threads. Like intrinsic metrics, it avoids needing a hand curated set of downstream targets, but like conglomerate benchmarks, it explicitly reasons about *many* tasks. Section 2.1 follows closely the recent work of Balestriero & LeCun (2024). Our framework also echoes several well-known results from the

classical theory of kernels. Notably, the trace term in Theorem 2.3 parallels the Hilbert–Schmidt Independence Criterion (HSIC) of Eq. (4) in (Gretton et al., 2007). Likewise, the term we get by taking the trace of $\mathbf{KG}$ is precisely the same as "kernel alignment" studied in the context of generalization (Cristianini et al., 2001), obtained by flattening each matrix and taking their inner product, as $\langle \mathbf{K}, \mathbf{G} \rangle = \mathrm{Tr}(\mathbf{KG})$.

There are some other related works that attack similar problems. In the computer vision space, VTAB (Zhai et al., 2020) takes a similar distributional view of tasks, but does not precisely characterize the distribution of tasks. Similar to our derivations, (Li et al., 2021) proposes a loss function based on the HSIC, which is an interesting avenue for future research.

## 5 DISCUSSION

Our results establish *Task Priors* as a principled method to avoid the costly and time-consuming practice of curating ever-larger suites of downstream benchmarks. By casting the space of tasks as a distribution over label graphs, we obtain closed-form expressions for the *expected* performance and its *variance* across all possible tasks, and we show that these statistics align with empirical linear probe performance on data from ImageNet.

### 5.1 LIMITATIONS AND FUTURE WORK

Despite these advances, several open issues remain. While the trace metrics correlate with probe accuracy, the correlation is not exact; closing this theory–practice gap will require a deeper empirical and theoretic study. Additionally, storing the full $n^2$ kernel can be prohibitive for very large datasets, although the matrices we observe are highly structured; further leveraging sparsity or sketching algorithms presents an immediate direction for further work. Our analysis is domain-agnostic, but our results in this paper focus on applications in learning representations for images. Further work is needed to examine its effectiveness on understanding the representations of Large Language Models (Skean et al., 2025), and, more generally, in natural language processing remains to be demonstrated. Additionally, this work focuses on a somewhat narrow definition of "task", in that we focus on classification. Additional study should check if a variety of other tasks, e.g. retrieval, ranking, generative, segmentation, also agrees with the Task Prior, or if there are other ways to adopt a distribution over theses tasks. This work also suggests, a prior-aware fine-tuning objective that simultaneously maximizes mean task performance while minimizing its variance, which remains an open avenue for future research. Tackling these questions can help lead to AI systems that perform *consistently well* across the vast landscape of tasks encountered in practice.

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

# A  MAIN THEORETICAL RESULTS

## A.1  PROOF OF THEOREM 2.3

*Proof.* The proof will involve a few different steps. First, we will solve the optimization objective $\mathcal{L}(\mathbf{W}, \mathbf{b}, \theta)$ for $\mathbf{b}$ and then for $\mathbf{W}$.

**Solving for $\mathbf{b}$.**  Let's first solve for the bias $\mathbf{b}$ directly since the loss is convex in $\mathbf{b}$

$$\nabla_{\mathbf{b}}\mathcal{L}(\mathbf{W}, \mathbf{b}, \theta) = \mathbf{0}$$

$$\iff \nabla_{\mathbf{b}}\frac{1}{N}\left\|\mathbf{W}f_\theta(\mathbf{X}) + \mathbf{b}\mathbf{1}_N^\top - \mathbf{Y}\right\|_F^2 = \mathbf{0}$$

$$\implies \frac{2}{N}\left(\mathbf{W}f_\theta(\mathbf{X}) + \mathbf{b}\mathbf{1}_N^\top - \mathbf{Y}\right)\mathbf{1}_N = \mathbf{0}$$

$$\iff \mathbf{W}f_\theta(\mathbf{X})\mathbf{1}_N + \mathbf{b}N - \mathbf{Y}\mathbf{1}_N = \mathbf{0}$$

$$\iff \mathbf{b} = \frac{1}{N}\mathbf{Y}\mathbf{1}_N - \frac{1}{N}\mathbf{W}f_\theta(\mathbf{X})\mathbf{1}_N.$$

From that we can simplify the loss by injecting the optimal value of the bias parameter as

$$\min_{\mathbf{b}}\mathcal{L}(\mathbf{W}, \mathbf{b}, \theta) = \min_{\mathbf{b}}\frac{1}{N}\left\|\mathbf{W}f_\theta(\mathbf{X}) + \mathbf{b}\mathbf{1}_N^\top - \mathbf{Y}\right\|_F^2$$

$$= \frac{1}{N}\left\|\mathbf{W}f_\theta(\mathbf{X}) + \left(\frac{1}{N}\mathbf{Y}\mathbf{1}_N - \frac{1}{N}\mathbf{W}f_\theta(\mathbf{X})\mathbf{1}_N\right)\mathbf{1}_N^\top - \mathbf{Y}\right\|_F^2$$

$$= \frac{1}{N}\left\|\mathbf{W}f_\theta(\mathbf{X})\mathbf{H} - \mathbf{Y}\mathbf{H}\right\|_F^2$$

with $\mathbf{H}$ the centering matrix $\mathbf{H} \triangleq \mathbf{I} - \frac{1}{N}\mathbf{1}_N\mathbf{1}_N^\top$.

**Solving for $\mathbf{W}$.**  Similarly to the derivation for $\mathbf{b}$, we can now optimize for $\mathbf{W}$ again using the fact that the loss is convex in $\mathbf{W}$ leading to the following

$$\nabla_{\mathbf{W}}\min_{\mathbf{b}}\mathcal{L}(\mathbf{W}, \mathbf{b}, \theta) = 0$$

$$\iff \nabla_{\mathbf{W}}\frac{1}{N}\left\|\mathbf{W}f_\theta(\mathbf{X})\mathbf{H} - \mathbf{Y}\mathbf{H}\right\|_F^2 = 0$$

$$\implies 2\frac{1}{N}\mathbf{W}f_\theta(\mathbf{X})\mathbf{H}f_\theta(\mathbf{X})^\top - 2\frac{1}{N}\mathbf{Y}\mathbf{H}f_\theta(\mathbf{X})^\top = 0$$

$$\iff \mathbf{W} = \mathbf{Y}\mathbf{H}f_\theta(\mathbf{X})^\top\left(f_\theta(\mathbf{X})\mathbf{H}f_\theta(\mathbf{X})^\top\right)^{-1}$$

from which we can again further simplify the loss as follows

$$\min_{\mathbf{W},\mathbf{b}}\mathcal{L}(\mathbf{W}, \mathbf{b}, \theta) = \min_{\mathbf{W}}\frac{1}{N}\left\|\mathbf{W}f_\theta(\mathbf{X})\mathbf{H} - \mathbf{Y}\mathbf{H}\right\|_F^2$$

$$= \frac{1}{N}\left\|\left(\mathbf{Y}\mathbf{H}f_\theta(\mathbf{X})^\top\right)\left(f_\theta(\mathbf{X})\mathbf{H}f_\theta(\mathbf{X})^\top\right)^{-1}f_\theta(\mathbf{X})\mathbf{H} - \mathbf{Y}\mathbf{H}\right\|_F^2$$

$$= \frac{1}{N}\left\|\mathbf{Y}\mathbf{V}\mathbf{S}^\top(\mathbf{S}\mathbf{S}^\top)^{-1}\mathbf{S}\mathbf{V}^\top - \mathbf{Y}\mathbf{H}\right\|_F^2$$

where we used the singular value decomposition $\mathbf{U}\mathbf{S}\mathbf{V}^\top$ of $f_\theta(\mathbf{X})\mathbf{H}$. We will now do some algebraic manipulations to simplify the above as follows

$$\min_{\mathbf{W},\mathbf{b}}\mathcal{L}(\mathbf{W}, \mathbf{b}, \theta) = \frac{1}{N}\left\|\mathbf{Y}\mathbf{H}\right\|_F^2 - 2\frac{1}{N}\operatorname{Tr}\left(\mathbf{H}^\top\mathbf{Y}^\top\mathbf{Y}\mathbf{V}\mathbf{S}^\top(\mathbf{S}\mathbf{S}^\top)^{-1}\mathbf{S}\mathbf{V}^\top\right)$$

$$+ \frac{1}{N}\operatorname{Tr}\left(\mathbf{V}^\top\mathbf{Y}^\top\mathbf{Y}\mathbf{V}\mathbf{S}^\top(\mathbf{S}\mathbf{S}^\top)^{-1}\mathbf{S}\right)$$

Now we use the fact that $\mathbf{V}^\top \mathbf{H} = \mathbf{V}^\top$ whenever the SVD was done on the centered $\mathbf{X}$ (which is the case here) to further simplify

$$
\begin{aligned}
\min_{\mathbf{W},\mathbf{b}} \mathcal{L}(\mathbf{W}, \mathbf{b}, \theta) =& \frac{1}{N} \|\mathbf{YH}\|_F^2 - 2\frac{1}{N} \operatorname{Tr}\left(\mathbf{V}^\top \mathbf{Y}^\top \mathbf{YV}(\mathbf{S}^2)^{-1}\mathbf{S}^2\right) \\
&+ \frac{1}{N} \operatorname{Tr}\left(\mathbf{V}^\top \mathbf{Y}^\top \mathbf{YV}(\mathbf{S}^2)^{-2}\mathbf{S}^4\right) \\
=& \frac{1}{N} \|\mathbf{YH}\|_F^2 - 2\frac{1}{N} \operatorname{Tr}\left(\mathbf{V}^\top \mathbf{Y}^\top \mathbf{YV}\right) + \frac{1}{N} \operatorname{Tr}\left(\mathbf{V}^\top \mathbf{Y}^\top \mathbf{YV}\right) \\
=& \frac{1}{N} \|\mathbf{YH}\|_F^2 - \frac{1}{N} \operatorname{Tr}\left(\mathbf{V}^\top \mathbf{Y}^\top \mathbf{YV}\right),
\end{aligned}
$$

as desired. We can minimize over $\theta$ on the left hand side and right hand side of the above equation and we get our result. $\qquad\square$

### A.2 PROOF OF LEMMA 2.5

*Proof.* Recall that $\mu(\mathbf{G}) = \frac{1}{Z_{T,\mathbf{K}}} e^{\frac{1}{T} \operatorname{Tr}(\mathbf{GK})}$. Then we can compute that,

$$
\begin{aligned}
\mathbb{P}(\mathbf{G}_{i,j} = 1) &= \mathbb{E}_\mu[\mathbf{G}_{ij}] \\
&= \frac{1}{Z_{T,\mathbf{K}}} \sum_{\mathbf{G}:\mathbf{G}_{i,j}=1} e^{\frac{1}{T} \operatorname{Tr}(\mathbf{GK})} \\
&= \frac{1}{Z_{T,\mathbf{K}}} \sum_{\mathbf{G}:\mathbf{G}_{i,j}=1} e^{\frac{1}{T} \sum_{i,j=1}^n \mathbf{G}_{i,j}\mathbf{K}_{i,j}} \\
&= \frac{1}{Z_{T,\mathbf{K}}} \sum_{\mathbf{G}:\mathbf{G}_{i,j}=1} \left[\prod_{1 \le k,l \le N} e^{\frac{1}{T}\mathbf{K}_{k,l}\mathbf{G}_{k,l}}\right].
\end{aligned}
$$

Now notice we can let:

$$
w_1 = \sum_{\mathbf{G}:\mathbf{G}_{i,j}=1} \left[\prod_{1 \le k,l \le N} e^{\frac{1}{T}\mathbf{K}_{k,l}\mathbf{G}_{k,l}}\right],
$$

$$
w_0 = \sum_{\mathbf{G}:\mathbf{G}_{i,j}=0} \left[\prod_{1 \le k,l \le N} e^{\frac{1}{T}\mathbf{K}_{k,l}\mathbf{G}_{k,l}}\right].
$$

And then,

$$
\mathbb{P}(\mathbf{G}_{i,j} = 1) = \frac{w_1}{w_0 + w_1}.
$$

Notice though that we can write,

$$
\sum_{\mathbf{G}:\mathbf{G}_{i,j}=1} \left[\prod_{1 \le k,l \le N} e^{\frac{1}{T}\mathbf{K}_{k,l}\mathbf{G}_{k,l}}\right] = e^{\frac{1}{T}\mathbf{K}_{i,j}} \sum_{\mathbf{G}:\mathbf{G}_{i,j}=0} \left[\prod_{1 \le k,l \le N} e^{\frac{1}{T}\mathbf{K}_{k,l}\mathbf{G}_{k,l}}\right].
$$

So,

$$
w_1 = e^{\frac{1}{T}\mathbf{K}_{i,j}} \cdot w_0.
$$

Then we know that,

$$
\mathbb{P}(\mathbf{G}_{i,j} = 1) = \frac{e^{\frac{1}{T}\mathbf{K}_{i,j}} \cdot w_0}{w_0 + e^{\frac{1}{T}\mathbf{K}_{i,j}} \cdot w_0} = \frac{e^{\frac{\mathbf{K}_{i,j}}{T}}}{1 + e^{\frac{\mathbf{K}_{i,j}}{T}}} = \sigma\left(\frac{\mathbf{K}_{i,k}}{T}\right).
$$

For the second part, we want to compute $\mathbb{P}(\mathbf{G}_{i,j}\mathbf{G}_{l,k} = 1)$, which we can note is clearly equivalent to $\mathbb{P}(\mathbf{G}_{i,j} = 1 \;\wedge\; \mathbf{G}_{l,k} = 1)$. As before, we can compute,

$$
\begin{aligned}
\mathbb{P}(\mathbf{G}_{i,j}\mathbf{G}_{l,k} = 1) &= \mathbb{E}_{\mu}[\mathbf{G}_{i,j}\mathbf{G}_{l,k}] \\
&= \frac{1}{Z_{T,\mathbf{K}}} \sum_{\mathbf{G}:\mathbf{G}_{i,j}\mathbf{G}_{l,k}=1} e^{\frac{1}{T}\operatorname{Tr}(\mathbf{G}\mathbf{K})} \\
&= \frac{1}{Z_{T,\mathbf{K}}} \sum_{\mathbf{G}:\mathbf{G}_{i,j}\mathbf{G}_{l,k}=1} e^{\frac{1}{T}\sum_{i,j=1}^{n}\mathbf{G}_{i,j}\mathbf{K}_{i,j}} \\
&= \frac{1}{Z_{T,\mathbf{K}}} \sum_{\mathbf{G}:\mathbf{G}_{i,j}\mathbf{G}_{l,k}=1} \left[ \prod_{1\le k,l\le N} e^{\frac{1}{T}\mathbf{K}_{k,l}\mathbf{G}_{k,l}} \right].
\end{aligned}
$$

We then let,

$$
w_1 = \sum_{\mathbf{G}:\mathbf{G}_{i,j}\mathbf{G}_{l,k}=1} \left[ \prod_{1\le k,l\le N} e^{\frac{1}{T}\mathbf{K}_{k,l}\mathbf{G}_{k,l}} \right],
$$

$$
w_0 = \sum_{\mathbf{G}:\mathbf{G}_{i,j}\mathbf{G}_{l,k}=0} \left[ \prod_{1\le k,l\le N} e^{\frac{1}{T}\mathbf{K}_{k,l}\mathbf{G}_{k,l}} \right].
$$

Which, we can note if $(i,j) \ne (l,k)$, then we have:

$$
\begin{aligned}
w_0 &= \sum_{\mathbf{G}:\mathbf{G}_{i,j}\mathbf{G}_{l,k}=0} \left[ \prod_{1\le k,l\le N} e^{\frac{1}{T}\mathbf{K}_{k,l}\mathbf{G}_{k,l}} \right] \\
&= \sum_{\mathbf{G}:\mathbf{G}_{i,j}=0,\mathbf{G}_{l,k}=0} \left[ \prod_{1\le k,l\le N} e^{\frac{1}{T}\mathbf{K}_{k,l}\mathbf{G}_{k,l}} \right] + \sum_{\mathbf{G}:\mathbf{G}_{i,j}=0,\mathbf{G}_{l,k}=1} \left[ \prod_{1\le k,l\le N} e^{\frac{1}{T}\mathbf{K}_{k,l}\mathbf{G}_{k,l}} \right] \\
&\quad + \sum_{\mathbf{G}:\mathbf{G}_{i,j}=1,\mathbf{G}_{l,k}=0} \left[ \prod_{1\le k,l\le N} e^{\frac{1}{T}\mathbf{K}_{k,l}\mathbf{G}_{k,l}} \right] \\
&= (e^{\frac{1}{T}(\mathbf{K}_{i,j}+\mathbf{K}_{l,k})})^{-1} \sum_{\mathbf{G}:\mathbf{G}_{i,j}=1,\mathbf{G}_{l,k}=1} \left[ \prod_{1\le k,l\le N} e^{\frac{1}{T}\mathbf{K}_{k,l},\mathbf{G}_{k,l}} \right] \\
&\quad + (e^{\frac{1}{T}\mathbf{K}_{i,j}})^{-1} \sum_{\mathbf{G}:\mathbf{G}_{i,j}=1,\mathbf{G}_{l,k}=1} \left[ \prod_{1\le k,l\le N} e^{\frac{1}{T}\mathbf{K}_{k,l}\mathbf{G}_{k,l}} \right] \\
&\quad + (e^{\frac{1}{T}\mathbf{K}_{l,k}})^{-1} \sum_{\mathbf{G}:\mathbf{G}_{i,j}=1,\mathbf{G}_{l,k}=1} \left[ \prod_{1\le k,l\le N} e^{\frac{1}{T}\mathbf{K}_{k,l}\mathbf{G}_{k,l}} \right] \\
&= (e^{-\frac{1}{T}(\mathbf{K}_{i,j}+\mathbf{K}_{l,k})} + e^{-\frac{1}{T}\mathbf{K}_{i,j}} + e^{-\frac{1}{T}\mathbf{K}_{l,k}}) \sum_{\mathbf{G}:\mathbf{G}_{i,j}=1,\mathbf{G}_{l,k}=1} \left[ \prod_{1\le k,l\le N} e^{\frac{1}{T}\mathbf{K}_{k,l}\mathbf{G}_{k,l}} \right] \\
&= (e^{-\frac{1}{T}(\mathbf{K}_{i,j}+\mathbf{K}_{l,k})} + e^{-\frac{1}{T}\mathbf{K}_{i,j}} + e^{-\frac{1}{T}\mathbf{K}_{l,k}})\, w_1.
\end{aligned}
$$

Then we can write that,

$$
w_0 + w_1 = (1 + e^{-\frac{1}{T}(\mathbf{K}_{i,j}+\mathbf{K}_{l,k})} + e^{-\frac{1}{T}\mathbf{K}_{i,j}} + e^{-\frac{1}{T}\mathbf{K}_{l,k}})\, w_1.
$$

So then,

$$\mathbb{P}(\mathbf{G}_{i,j}\mathbf{G}_{l,k} = 1) = \frac{w_1}{w_0 + w_1}$$

$$= \frac{1}{(1 + e^{-\frac{1}{T}(\mathbf{K}_{i,j}+\mathbf{K}_{l,k})} + e^{-\frac{1}{T}\mathbf{K}_{i,j}} + e^{-\frac{1}{T}\mathbf{K}_{l,k}})}$$

$$= \frac{1}{(1 + e^{-\frac{1}{T}\mathbf{K}_{i,j}})(1 + e^{-\frac{1}{T}\mathbf{K}_{l,k}})}$$

$$= \sigma(\frac{\mathbf{K}_{i,j}}{T})\sigma(\frac{\mathbf{K}_{l,k}}{T}).$$

$\square$

### A.3 PROOF OF THEOREM 2.6

*Proof.* The first equality in the equation follows from the linearity of expectation, and the characterization that,

$$\mathrm{Tr}(\mathbf{MG}) = \sum_{1 \leq i,j \leq N} \mathbf{M}_{i,j}\mathbf{G}_{i,j},$$

for $\mathbf{M}, \mathbf{G}$ symmetric matrices. Then, notice that this is a weighted sum of independent Bernoulli random variables. So, $\mathbb{E}_{\mathbf{G}\sim\mu_{\mathbf{K}}}[\mathbf{G}_{i,j}] = \mathbb{P}_{\mathbf{G}\sim\mu_{\mathbf{K}}}(\mathbf{G}_{i,j} = 1)$ and we can apply the above lemma and we are done.

For the second part, since this is a sum of independent random variables, we may use,

$$\mathrm{Var}(\mathrm{Tr}(\mathbf{MG})) = \sum_{1 \leq i,j \leq N} \mathbf{M}_{i,j}^2 \, \mathrm{Var}(G_{i,j}) \tag{8}$$

$$= \sum_{1 \leq i,j \leq N} \mathbf{M}_{i,j}^2 \mathbb{P}(\mathbf{G}_{i,j} = 1)(1 - \mathbb{P}(\mathbf{G}_{i,j} = 1)) \tag{9}$$

$$= \sum_{1 \leq i,j \leq N} \mathbf{M}_{i,j}^2 \sigma(\mathbf{K}_{i,j}/T)(1 - \sigma(\mathbf{K}_{i,j}/T)). \tag{10}$$

$\square$

# B ADDITIONAL EMPIRICAL STUDIES

## B.1 ABLATION ON CHOICE OF KERNEL

In Figure 6, we can see how the choice of task prior kernel matrix affects the downstream computation of the mean and variance of $\mathrm{Tr}(\mathbf{MG})$. As we might expect, we see that generally the mean $\mathbb{E}_{\mathbf{G} \sim \mu_{\mathbf{K}}}[\mathrm{Tr}(\mathbf{MG})]$ is higher when the task prior kernel $\mathbf{K}$ matrix is the same as the the matrix being evaluated $\mathbf{M}$. We can also see that models in the same family tend to demonstrate higher

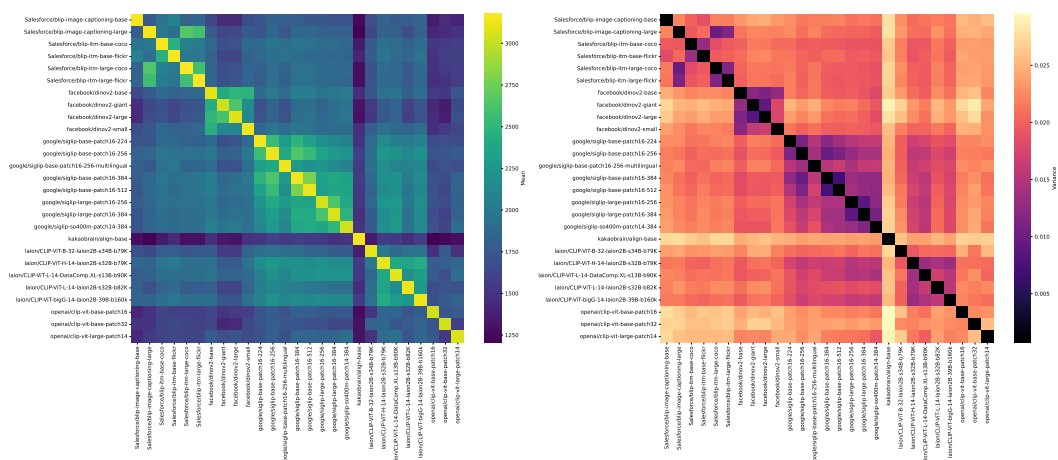

Figure 6: Here, we show a comparison of how the choice of task prior kernel $\mathbf{K}$, reflected here in the color of the data points, affects the the evaluation of the mean and variance of $\mathrm{Tr}(\mathbf{MG})$. Each point is computed via the exact formulas given in 2.6, with a temperature of $T = 0.01$.

## B.2 ABLATION ON THE TEMPERATURE PARAMETER

In Figure 7, we can see the effect of the sampler changing the temperature in the measure. We can see how increasing temperature increases diversity but also brings us closer to a uniform distribution over labels.

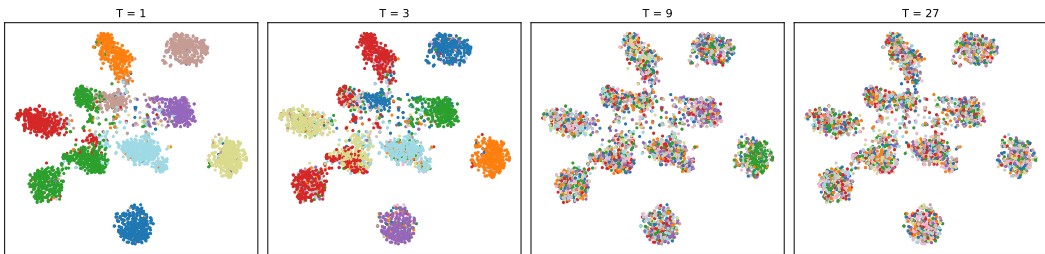

Figure 7: We show a TSNE plot of Imagenette, with labels generated by the sampling Algorithm 1 for four choices of temperature.

### B.3 EVALUATION OF *timm* MODELS WITH TASK PRIORS

Using our sampling algorithm, we draw tasks from a prior $\mu_{\mathbf{K}}$ induced by *efficientnet_b5*. For each of 33 models from *timm* Wightman (2019), we train an independent linear probe on every sampled task with *efficientnet_b5* and record the resulting accuracies. We then compare this to simply computing $\mathbb{E}_{\mu_{\mathbf{K}}}[\mathrm{Tr}(\mathbf{MG})]$ and $\mathrm{Var}_{\mu_{\mathbf{K}}}(\mathrm{Tr}(\mathbf{MG}))$ as per 2.6.

We report the results of this study in Figure 8, where we find that models that perform better on average also tend to have a better variance over tasks.

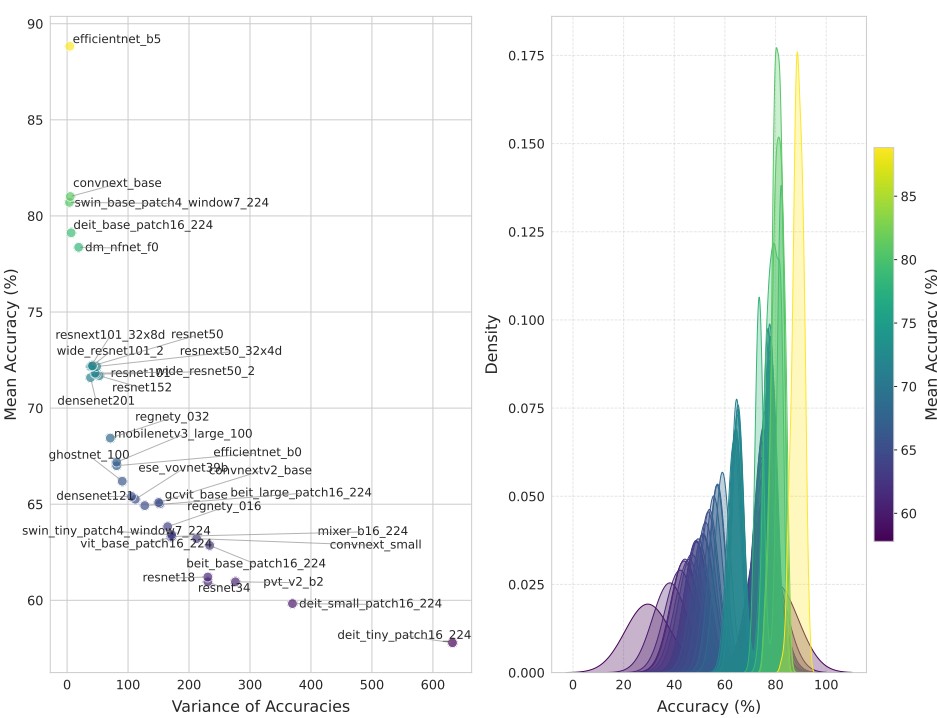

Figure 8: Linear probe performance of a selection of models from *timm* on a distribution of binary labels sampled by the *task prior* on the mini-Imagenet dataset. Here we use *efficientnet_b5* as the backbone model for the task prior distribution.

## C CODE AND REPRODUCIBILITY

All code to reproduce the central results of this paper are available at the following anonymized repository: https://anonymous.4open.science/r/taskpriors-526A/

