# OpenReview forum: "Task Priors: Enhancing Model Evaluation by Considering the Entire Space of Downstream Tasks"
_ICLR.cc/2026/Conference — Submitted to ICLR 2026_

### Official Review · Reviewer_rciQ · 2025-10-28

**Soundness:** 2
**Presentation:** 2
**Contribution:** 3
**Rating:** 4
**Confidence:** 4

**Summary:**

The paper tackles the problem of evaluating pre-trained backbone models on a variety of downstream tasks. It addresses the limitation of current evaluation protocols that rely on a small, fixed set of benchmarks, which fail to capture the full range of real-world applications. The paper proposes a proxy evaluation framework that models downstream tasks as samples from a probability space — modelling a Task Prior distribution based on a feature kernel to compute expectation and variance of the model performance without retraining new classifiers or build more benchmarks.

From my understanding, the authors frame downstream tasks as training linear classifiers on top of pre-trained representations $f(X)$ to predict target variables. Tasks are defined by the label graph $G = Y^{T}Y$, where $Y$ represents the target labels of a specific dataset. Each label graph $G$ is treated as a task, and the space of tasks is represented as a collection of $G$ matrices, weighted by the feature kernel $K$ derived from $f(X)$. A Gibbs distribution is imposed on this task space, serving as the Task Prior, which facilitates natural inference and sampling based on established theoretical results. The performance of a task is quantified as the loss metric evaluated based on Eq. (2). By applying the Gibbs distribution over the space of label graphs, the authors derive tractable expressions for the expected performance and variance of performance from using the pre-trained representations.

**Strengths:**

The idea of tractable evaluation of model performance on diverse new tasks is very interesting and significantly useful to me. The method is elegantly simple with strong theoretical motivation and empirical results to substantiate the proposed claims.

**Weaknesses:**

1) While the introduction is clear, I find the presentation of the main results in Section 2 is extremely difficult to follow, which requires me to go back and forth multiple times to finally understand the definition of tasks and performance as well as the motivation of using the label graphs and feature kernels.

2) The paper would have been easier to follow if the scope of the tasks and the key concepts (as mentioned above) could be explicitly explained somewhere at the beginning.

3) Furthermore, the task definition in this work is limited to tasks whose target variables lie within the same space and on a fixed input dataset. If it is a classification task, the formulation requires the same number of classes.

4) For a large dataset, the computation of the kernel matrix is incredible challenging, which hinders applicability of the proposed approach, yet remains undiscussed in the current paper.

**Questions:**

1) Can the framework model diverse classification tasks across different domains (e.g., image classification vs. visual question answering with multiple-choice settings) together, or is it limited to tasks with similar characteristics?

2) How are the label graph and feature kernel computed efficiently for large datasets in practice?

3) Could the proposed method handle more complex modality such as texts and videos?

4) In Theorem 2.6 (lines 216-226), how is this additional kernel matrix $M$ defined and designed? Can any matrix based on any similarity metric or kernel be used for the computation, or what properties should $M$ satisfy to be used in Eq. (6)?

---

> ### Author Response · Authors · 2025-11-21
>
> We would firstly like to thank the reviewer for their time and thoughtful review of our work, which has helped us improve our paper. We address your comments and concerns below:
>
> > The paper would have been easier to follow if the scope of the tasks and the key concepts (as mentioned above) could be explicitly explained somewhere at the beginning.
>
> Thank you for this note, we have added text at the begining of this section to help with readability.
>
> >  If it is a classification task, the formulation requires the same number of classes.
>
> Since we evaluate our models on a distribution of tasks, if we have a representations that have a linear probe that performs very well on, n-class classification for a variety of choices of labels, then these representations are able to also perform well on m-class settings where m>n.
>
> > For a large dataset, the computation of the kernel matrix is incredible challenging, which hinders applicability of the proposed approach, yet remains undiscussed in the current paper. / How are the label graph and feature kernel computed efficiently for large datasets in practice?
>
> In general with this method we do not need to compute the whole kernel matrix. We present two areas you may need to use the kernel matrix. One is in Algorithm 1, which we use to sample the downstream tasks, as well as in Theorem 2.6, where we immediatly compute the expectation and variance.
>
> To address the first, our algorithm only requires us to hold in memory low-rank factors of our kernel matrix. So in general this could will require us to store $2nr$ floats, for rank $r$ factors, which is much better than the $n^2$ mentioned. This is a natural assumption, for instance you could be using a linear or cosine kernel compute on representations in $\mathbb R^d$, then this means we only need to hold in memory $O(nd)$ terms.
>
> To address the second, we note that these formulas for both the expectation and variance consist of a sum over $i,j$. This means that to compute this we could only store at any one point in time the kernel entry $\mathbf K_{i,j}$, and sum the scalers over this.
>
> > Can the framework model diverse classification tasks across different domains?
>
> Though we do not do these experiments in the paper, the framework is not nesesarily restricted to 'homogeneous' tasks; it operates on any label graphs over an arbitrary finite set of examples in a shared representation space. As long as different tasks can be expressed on the same feature space, they can be modeled by a Task Prior. In particular, multi-choice VQA can be cast as classification over candidate answers for each (image, question) pair, so its labelings fit directly into our label-graph formalism. In that case the graph would have a connection only between representations in the same (image, question) pair.
>
> > Could the proposed method handle more complex modality such as texts and videos?
>
> Yes, our architecture is entirely domain agnostic, and should in principle generalize well to videos and text, which present very interesting directions for future research.
>
> > In Theorem 2.6 (lines 216-226), how is this additional kernel matrix M defined and designed? Can any matrix based on any similarity metric or kernel be used for the computation, or what properties should satisfy to be used in Eq. (6)?
>
> Here the $\mathbf M$ is the kernel matrix generated by each of the models that we are trying to evaluate, and $\mathbf K$  is the kernel matrix for the model we are aiming to evaluate. We have added additional text in the paper to better explain the difference between $\mathbf M$ and $\mathbf K$. In that sense, all we need is that $\mathbf M$ is a valid kernel matrix, and is the same size as $\mathbf K$. In practice we take $\mathbf M$ to be the cosine similarity kernel from the representations we aim to evaluate.

---

> > ### Comment · Reviewer_rciQ · 2025-11-25
> >
> > Thank you for the responses. The authors have addressed most of my concerns. While I am not very impressed with the technical part and the novelty of the theoretical results is rather limited, I think it is addressing a problem of timely importance. The paper can provide a good foundation for future works to build on for addressing more complex settings. Hence, I am willing to increase the score.
> >
> > However, some more clarifications are needed in your responses for completeness.
> >
> > 1. The authors claim: *"if we have a representations that have a linear probe that performs very well on, n-class classification for a variety of choices of labels, then these representations are able to also perform well on m-class settings where m>n."*
> > I doubt whether this certainly holds. Can it be proven?
> >
> > 2. In the 3rd response, what is Theorem X?
> >
> > 3. Finally, please seriously consider improving the presentation per my comments. And make sure to precisely clarify in the paper the diversity level of the tasks your method can handle. This would be greatly informative for future research.

---

> ### Author Response · Authors · 2025-11-27
>
> Thank you for agreeing to raise your score! To respond to your other comments.
>
> 1. Indeed it can be proven in some settings. For instance, if you had 8 data points and you are able to classify both of these binary labels with linear classifier, then it is clear that you can also classify the third set of labels, where there are four classes.
>
> | x1 | x2 | x3 | x4 | x5 | x6 | x7 | x8 |
> |-------|-------|-------|-------|-------|-------|-------|-------|
> | 0  | 0  | 0  | 0  | 1  | 1  | 1  | 1  |
> | 1  | 1  | 0  | 0  | 1  | 1  | 0  | 0  |
> | 1  | 1  | 2  | 2  | 3  | 3  | 4  | 4  |
>
> 2. Sorry for the editorial issue, this refers to Theorem 2.6.
>
> 3. We have updated the paper to include a description of each section to improve readability. We will also further improve the work for the camera ready version to improve clarity.

---

### Official Review · Reviewer_RNUm · 2025-10-28

**Soundness:** 3
**Presentation:** 2
**Contribution:** 3
**Rating:** 6
**Confidence:** 3

**Summary:**

The paper introduces Task Priors, a novel framework for evaluating pretrained models by defining a probability distribution over downstream tasks, rather than relying on fixed benchmark datasets. It defines a Gibbs distribution over label graphs that captures the likelihood of different downstream tasks based on kernel similarity, offering a probabilistic view of evaluation. Empirical applications validate the proposed metrics.

**Strengths:**

- The paper redefines model evaluation as a probabilistic process over all possible tasks, providing a fresh and mathematically principled alternative to the static-benchmark paradigm.
- Strong theory that establishes a connection between supervised and self-supervised objectives via kernel alignment and trace formulations.
- The framework unifies representation-based and task-based evaluation, potentially becoming a standard for fair model comparison.

**Weaknesses:**

- Although the method is domain-agnostic in principle, experiments and formulations are focused solely on classification; it’s unclear how Task Priors extend to retrieval, regression, or generative tasks.
- The framework heavily relies on the definition of kernel similarity. Sensitivity to kernel type, temperature $T$, and feature normalization could affect reliability.
- No guarantee is provided that Task Priors predict performance in unseen domains.

**Questions:**

See the above weaknesses.

---

> ### Author Response · Authors · 2025-11-21
>
> We would firstly like to thank the reviewer for their time and thoughtful review of our work, which has helped us improve our paper. We address your comments and concerns below:
>
>
> > it’s unclear how Task Priors extend to retrieval, regression, or generative tasks.
>
> Our theory already covers multi-class classification via label graphs, so it is not limited to binary tasks. More broadly, we view classification as a rich primitive. If a representation supports good performance on a wide range of classification problems, then standard architectures (for instance small MLP heads) should be able to realize common regression targets as functions of those features. Likewise, some retrieval problems can be cast as n-way classification over a candidate set for each query. In this paper we therefore focus on classification tasks, both because they fit the theory exactly and because they can serve as a practical proxy for the broader capabilities mentioned above. We will add text to the revision to make this more clear.
>
> > The framework heavily relies on the definition of kernel similarity.
>
> We agree the kernel is a key modeling choice, and our framework is designed to allow the practitioner to pick a kernel which wil correspond to their inductive bias about which tasks are most likely to oberve. In practice we use normalized features and cosine similarity, so scaling of the kernel is mostly absorbed into the temperature, and we showcase how changing the temperature effects the sampled labels in Appendix Figure B.2.
>
> > No guarantee is provided that Task Priors predict performance in unseen domains.
>
> Our goal in this work is empirical study of the performance of models across a wide distribution of tasks rather than to prove generalization guarantees under arbitrary domain shift. We show that Task Prior statistics correlate well with average linear-probe performance over many tasks in the MIEB. Notably, we only use data from ImageNet to predict this performance, whereas MIEB utlizes a wide array of datasets.

---

### Official Review · Reviewer_JFBG · 2025-10-31

**Soundness:** 2
**Presentation:** 2
**Contribution:** 2
**Rating:** 4
**Confidence:** 3

**Summary:**

The paper proposes a new framework that moves beyond traditional representation learning evaluation approaches. Rather than assessing models on a fixed set of downstream tasks, the authors introduce the concept of task priors, a probabilistic representation of the space of all possible downstream tasks that a model might encounter.

**Strengths:**

1. Significance: This work, if clarity issues mentioned below are resolved, can have a broad impact on representation learning so that the standard linear probing evaluation protocol becomes unnecessary.
2. Originality is hard to evaluate due to limited domain expertise, see clarity and quality concerns below.

**Weaknesses:**

1. Motivation: I wonder why training linear probes for evaluation is a computational bottleneck that we have to avoid since linear probes are quite cheap to train (compared to the time to build the representation) and often done in a few shot manner.
2. There are a few clarity issues about the methodology that make it hard to evaluate the quality of this paper, see questions below.

**Questions:**

1. Can you add legend to Figure 1 and explain how 9 smaller figures are the right probabilistic?
2. [Important] I wonder if authors can further give examples for what is G and what is K in the standard feature representation (supervised at least) learning context (e.g., do we need labels for downstream tasks? If yes, we still need the manual collection of downstream datasets.), and which G and K are used to generate each datapoints in Figure 4. This can help readers understand your procedure better and how to use your method in practice, especially given that you have almost 1 more page. Besides, please double check the paper and use the fixed notation for matrices such as $\mathbf{G}$ instead of $G$.
3. For Theorem 2.6, why do we introduce this new kernel matrix M and what is its relationship with K? Why is Tr(MG) related to the linear classifier performance, not Tr(KG)? Also please fix typos for theorem references (line 140-141).
4. What’s the relationship between V and Z? In 293, can you explain why we can achieve a speedup?
5. Can you explain how Algorithm 1 is used exactly to compute the mean and variance term after sampling labels?
6. Can you compare the time you need to compute linear probes for each dataset vs. time using your method?
7. Can you explain the relationship of your paper to “Provable guarantees for self-supervised deep learning with spectral contrastive loss” by Haochen et al. (2021), which proposes a graph-based spectral loss and provides theoretical analysis based on linear probe generalization error?
8. Seems like this is one important assumption: “the performance on a hand-curated collection of downstream tasks should follow the distribution implied by the Task Prior.” How bad can the estimation be if this assumption is violated and how to enforce it in practice?

---

> ### Author Response · Authors · 2025-11-21
>
> We would firstly like to thank the reviewer for their time and thoughtful review of our work, which has helped us improve our paper. We address your comments and concerns below:
>
> > I wonder why training linear probes for evaluation is a computational bottleneck that we have to avoid since linear probes are quite cheap to train (compared to the time to build the representation) and often done in a few shot manner.
>
> We actually are primarily trying to address the research bottleneck that comes from not having a strong distribution of target labels to evaluate our model on.
>
> > Can you add legend to Figure 1 and explain how 9 smaller figures are the right probabilistic?
>
> The colors in this figure correspond to the class label. We have added a description to this Figure 1. The 9 smaller figures are sampled labels using Algorithm 1. These are $9$ samples from this probability distribution, and with our method we are able to find many such samples.
>
> > I wonder if authors can further give examples for what is G and what is K in the standard feature representation (supervised at least) learning context (e.g., do we need labels for downstream tasks? If yes, we still need the manual collection of downstream datasets.), and which G and K are used to generate each datapoints in Figure 4.
> For Theorem 2.6, why do we introduce this new kernel matrix M and what is its relationship with K? Why is Tr(MG) related to the linear classifier performance, not Tr(KG)? Also please fix typos for theorem references (line 140-141).
>
> In our pipeline, we will typicall take $\mathbf K$ to be the kernel matrix generated by cosine similarity for feature representation $f_\theta(\mathbf X)$. Generally, $\mathbf G$ is an adjacency matrix graph on the data points. In some settings of our paper we take this graph to be generated by labels, where the two points are connected if they are in the same class. However we will typically take $\mathbf G$ to be *probabilistic.* In that sense, we have a probability distribution over $\mathbf G$, $\mu_\mathbf{K}(\mathbf G)$, which is generated only by the kernel matrix. In this sense we do **not** need access to manually collected labels for downstream tasks.
>
> In Figure 4, we are comparing the accuracy we see over linear probes on tasks which are sampled from our task prior, to the direct computation of $\mathbb E_{\mu_\mathbf{K}}[\text{Tr}(\mathbf{MG})]$, and the associated variance. Here the $\mathbf M$ is the kernel matrix generated by each of the models that we are trying to evaluate, and $\mathbf K$  is the kernel matrix for the model we are aiming to evaluate. We have added additional text in the paper to better explain the difference between $\mathbf M$ and $\mathbf K$.
>
> Our theory shows that, if we had access to set of labels and associated label graph, that the performance of the representations $f(\mathbf X)$ is related to $\text{Tr}(\mathbf {MG})$, if $\mathbf M$ is the kernel generated by $f(\mathbf X)$. However this requires access to a set of labels, which we assume we will not have access to in general.
>
> > Besides, please double check the paper and use the fixed notation for matrices
>
> Thank you for this note, we have fixed this in the updated version of the paper.
>
> > What’s the relationship between V and Z? In 293, can you explain why we can achieve a speedup?
>
> In Algorithm 1, $\mathbf Z$ represents a potentially low-rank factor of the kernel matrix $\mathbf K$, which will always exist since $\mathbf K$ must be a positive semi-definite matrix. In our preceding theory, specifically in Theorem 2.3, $\mathbf V$ denotes the right singular vectors of $f(\mathbf X)$. So, $\mathbf{VV^T}$ is a valid kernel, and $\mathbf V$ is therefore a valid choice of low-rank factor $\mathbf Z$. However our algorithm is more general, and allows for other choices of kernel matrix $\mathbf K$, hence the change in notation.
>
> > Can you explain how Algorithm 1 is used exactly to compute the mean and variance term after sampling labels?
>
> We have two related methods which we compare in the paper. For one, we can use Algorithm 1 to sample classification tasks, and then fine tune linear probes on these tasks and report these accuracies. On the other hand, we also offer a method that allows us to directly compute $\mathbb E_{\mu_\mathbf{K}}[\text{Tr}(\mathbf{MG})]$ and $\text{Var} _{\mu_\mathbf{K}}[\text{Tr}(\mathbf{MG})]$, which offers another way to compare performance of representations with respect to a prior distribution.
>
> > Can you compare the time you need to compute linear probes for each dataset vs. time using your method?
>
> Even for only 4 models, it takes 37.8 seconds to compute linear probes and 0.635 seconds to compute the task prior expectation and variance on a single A100 gpu.

---

> ### Author Response · Authors · 2025-11-21
>
> (cont. from before)
>
> > Can you explain the relationship of your paper to “Provable guarantees for self-supervised deep learning with spectral contrastive loss” by Haochen et al. (2021), which proposes a graph-based spectral loss and provides theoretical analysis based on linear probe generalization error?
>
> Thank you for the suggestion. Haochen et al. (2021) and our work are related at a conceptual level, in that both analyze self-supervised representations through a kernel/graph alignment view and connect this to linear probe performance. In these sense our results corroborate theirs. Their focus, however, is on a specific training objective, and they provide guarantees for a single underlying ground-truth task.
>
> In contrast, our paper takes a fixed representation and develops a probabilistic Task Prior over all possible downstream tasks on a dataset. We then derive closed-form expressions for the average and variability of linear probe performance across this task distribution. Thus, while both works use a similar mathematical lens, Haochen et al. study how a particular objective leads to good features, whereas we study how a given representation behaves across an ensemble of downstream tasks.
>
> > Seems like this is one important assumption: “the performance on a hand-curated collection of downstream tasks should follow the distribution implied by the Task Prior.” How bad can the estimation be if this assumption is violated and how to enforce it in practice?
>
> We agree that it is not immediately obvious that the tasks that people in the community care about are those implied by the task prior. This is a priori not an assumption, but rather a claim which we wish to test empirically. We do this in Figure. 5, where we see that we observe high correlation ~0.82 and ~0.74 between the curated benchmark of downstream tasks and the statistics computed based on the kernel matrix.

---

### Official Review · Reviewer_jKi3 · 2025-11-01

**Soundness:** 3
**Presentation:** 3
**Contribution:** 3
**Rating:** 4
**Confidence:** 3

**Summary:**

Current model evaluation relies on finite, hand-curated benchmark suites (e.g., ImageNet, GLUE, MTEB), which capture only a small subset of real-world tasks. As the diversity of downstream applications grows, this fixed-benchmark paradigm forms a structural bottleneck for assessing generalization. The paper proposes a probabilistic framework to replace discrete benchmarks with continuous task distributions. To achieve this, the authors show that absolute evaluation metrics, such as classification accuracy, are equivalent to relative evaluation metrics (sample similarity). For the relative metric, the knowledge of the similarity graph suffices for evaluation. So the authors define downstreams tasks by similarity graphs and propose a way to define distributions over tasks as a distribution over graphs. Given this "task prior", the authors provide efficient ways to evaluate the mean performance and the variance of performance of any given representation. The authors also provide a method for sampling downstream tasks from the prior and an SSL-inspired approach to defining task priors.

**Strengths:**

The proposed method connects supervised, self-supervised, and kernel-alignment evaluations under a single theoretical lens via Theorem 2.3. The authors provide efficient methods for evaluating the mean and variance of downstream performance with respect to the proposed task prior.

The paper provides good empirical coverage of experiments on diverse architectures (CLIP, SigLIP, BLIP, DinoV2). The experiments show a strong correlation between the calculated metric and the linear probe, and with real benchmark outcomes.

**Weaknesses:**

The core mathematics (e.g., Gibbs measure, kernel alignment, HSIC) builds directly on established theory, albeit with a fresh interpretation for model evaluation. The effect of using different temperatures or selecting the optimal temperature is not described in detail. It would be great to provide a sequence of assumptions that motivate the effectiveness of using a Gibbs distribution to define the task prior. Currently, the prior task is very heuristic, as it is motivated by a metric that should be correlated with the mean of downstream performance. It is unclear why the variance calculated under the proposed task prior should be indicative of the variance in performance across downstream tasks.

The need to select a kernel for the task prior shifts the problem without addressing it. Results can vary depending on which model’s kernel is used as the Task Prior (e.g., DinoV2 vs. SigLIP). The implications of this bias are not fully explored. The SSL kernel does not seem very useful on its own. If we end up using a large foundation model to define the kernel, we can directly compare representation similarities by checking kernel similarities between representations. The task prior distribution (Gibbs distribution) is not very well motivated.

Does the additional step of defining the task before computing expectations improve the correlation with downstream empirical performance?

The framework also needs access to full kernel matrices (O($n^2$) memory), which becomes impractical for very large datasets.

**Questions:**

Please refer to the Weaknesses section.

---

> ### Author Response · Authors · 2025-11-21
>
> We would firstly like to thank the reviewer for their time and thoughtful review of our work, which has helped us improve our paper. We address your comments and concerns below:
>
> > It would be great to provide a sequence of assumptions that motivate the effectiveness of using a Gibbs distribution to define the task prior.
>
> Using a Gibbs distribution for the Task Prior is primarily a decision made out of nessesity. The ability for us to factor the Gibbs distribution as in Lemma 2.5 and Theorem 2.6 is nessesary for us to derive our important equations, which enable our algorithms to hold. If we had for instance, defined our distribution to be proportional to $\text{Tr} (\mathbf G \mathbf K)$, then these formula would not have been tractable.
>
> This is also a very natural choice of distribution, as it corresponds to applying the `softmax` function across all the label-graphs.
>
> > It is unclear why the variance calculated under the proposed task prior should be indicative of the variance in performance across downstream tasks.
>
> In Theorem X we show that the optimum for both the best linear probe on $Y$ (in terms of MSE) is equivalent to $\text{Tr}(\mathbf G \mathbf K)$, for a specific choice of $\mathbf K$ given by the SVD of the features, and where $\mathbf G = \mathbf Y^T \mathbf Y$. In that sense, the variance in true performance accross tasks $Y$ should also be represented in the variance of $\text{Tr}(\mathbf G \mathbf K)$ across $\mathbf G$. However, this theoretical note is different from the way we measure this in practice, so we verify this correspondance emperically in Figure 4.
>
> > The need to select a kernel for the task prior shifts the problem without addressing it.
>
> The primary problem is that there are many domains where we want to have models that learn strong representations but we do not have a strong coverage over downstream tasks for practitioners to be able to develop these models. In that sense our work here addresses this problem. For a practitioner, this could mean using the older version of your model while you develop your next model, or as you mention using other inductive biases such as SSL objectives to define a prior.
>
> Future work is needed to better understand how to specify priors, and how to construct them even in the absence of a fixed backbone foundation model. These directions build conceptually on the framework introduced in this paper, so we believe establishing and validating this formulation is a natural prerequisite for this kind of subsequent work.
>
> > The framework also needs access to full kernel matrices (O(n^2) memory), which becomes impractical for very large datasets.
>
> We present two areas you may need to use the kernel matrix. One is in Algorithm 1, which we use to sample the downstream tasks, as well as in Theorem 2.6, where we immediately compute the expectation and variance.
>
> To address the first, our algorithm only requires us to hold in memory low-rank factors of our kernel matrix. So in general this could will require us to store $2nr$ floats, for rank $r$ factors, which is much better than the $n^2$ mentioned. This is a natural assumption, for instance you could be using a linear or cosine kernel compute on representations in $\mathbb R^d$, then this means we only need to hold in memory $O(nd)$ terms.
>
> To address the second, we note that these formulas for both the expectation and variance consist of a sum over $i,j$. This means that to compute this we could only store at any one point in time the kernel entry $\mathbf K_{i,j}$, and sum the scalers over this.
>
> > Does the additional step of defining the task before computing expectations improve the correlation with downstream empirical performance?
>
> Do you mean test if we skew the task prior distribution in some way towards tasks we have observed? We do not test this in the paper but it is an interesting area for future research.

---

### Meta-Review · Area_Chair_FRPo · 2025-12-14

**Summary:**

This paper introduces Task Priors, a novel probabilistic framework that models the space of all downstream tasks as label graphs $G$ via a Gibbs distribution weighted by a feature kernel $K$ to compute the mean performance and variance of performance of pre-trained representations efficiently.
However, the primary flaws identified during the review process, which remain significant concerns, include:

1.Limited Novelty and Motivation: The core theory relies on established results e.g., kernel alignment, Gibbs measure, and the motivation for using the specific Gibbs distribution for the Task Prior.

2.Kernel Selection and Bias: The framework shifts the problem to selecting a kernel $K$, or $M$ for evaluation, introducing bias.

3.Scalability Concerns: The original approach relies on full kernel matrices, which is $O(n^2)$ memory and impractical for very large datasets, though the authors propose mitigating this via low-rank factors $O(nd)$ or $O(nr)$ memory.

Despite the authors' responses, the fundamental concerns regarding the novelty, theoretical motivation of the Task Prior, and clarity of the presentation persisted.

**Reviewer Concerns:**

The following core concerns were a major factor in the final ratings:

1.Weak Theoretical Motivation: The choice and justification of the Gibbs distribution for the Task Prior and the reliability of the resulting variance calculation remained unconvincing.

2.Kernel Dependence: The reliance on selecting a kernel $K$ for the prior was seen as an unsolved problem that introduces bias.

3.Presentation: The paper's core methodology was criticized as extremely difficult to understand, hindering evaluation quality.

**Reviewer Scores:**

Reviewer rciQ shows the sign of partially increasing the score, while I hardly see other reviewers' score change signals after checking the rebuttal and the updated manuscript.

---

### Decision · Program_Chairs · 2026-01-26

Reject